# A Novel Cuffless Blood Pressure Prediction: Uncovering New Features and New Hybrid ML Models

**DOI:** 10.3390/diagnostics13071278

**Published:** 2023-03-28

**Authors:** Majid Nour, Kemal Polat, Ümit Şentürk, Murat Arıcan

**Affiliations:** 1Department of Electrical and Computer Engineering, Faculty of Engineering, King Abdulaziz University, Jeddah 21589, Saudi Arabia; 2Department of Electrical and Electronics Engineering, Faculty of Engineering, Bolu Abant Izzet Baysal University, 14280 Bolu, Turkey; 3Department of Computer Engineering, Faculty of Engineering, Bolu Abant Izzet Baysal University, 14280 Bolu, Turkey

**Keywords:** hypertension, PPG, blood pressure prediction, cuffless blood pressure, regression, more accurate models

## Abstract

This paper investigates new feature extraction and regression methods for predicting cuffless blood pressure from PPG signals. Cuffless blood pressure is a technology that measures blood pressure without needing a cuff. This technology can be used in various medical applications, including home health monitoring, clinical uses, and portable devices. The new feature extraction method involves extracting meaningful features (time and chaotic features) from the PPG signals in the prediction of systolic blood pressure (SBP) and diastolic blood pressure (DBP) values. These extracted features are then used as inputs to regression models, which are used to predict cuffless blood pressure. The regression model performances were evaluated using root mean squared error (RMSE), R^2^, mean square error (MSE), and the mean absolute error (MAE). The obtained RMSE was 4.277 for systolic blood pressure (SBP) values using the Matérn 5/2 Gaussian process regression model. The obtained RMSE was 2.303 for diastolic blood pressure (DBP) values using the rational quadratic Gaussian process regression model. The results of this study have shown that the proposed feature extraction and regression models can predict cuffless blood pressure with reasonable accuracy. This study provides a novel approach for predicting cuffless blood pressure and can be used to develop more accurate models in the future.

## 1. Introduction

Hypertension, a severe medical condition that significantly increases the risk of heart attack, stroke, kidney failure, and blindness and is the leading cause of premature death worldwide, affects an estimated 1.13 billion people, with fewer than one in five having it under control [1]. High blood pressure is responsible for approximately 54% of strokes and 47% of coronary heart diseases worldwide [2]. It is estimated that one in three adults has hypertension, and the prevalence is even higher among specific populations, such as African Americans and the elderly. This essay will discuss the epidemiology of hypertension mortality, the risk factors associated with hypertension mortality, and the strategies for its prevention. It will also explore the implications of hypertension mortality on public health.

Hypertension, commonly known as high blood pressure, is a condition in which the force of the blood against the artery walls is consistently too high. This increased pressure can have serious and potentially deadly consequences, as it can damage the arteries, the heart, and other organs in the body. It is a major risk factor for heart attack, stroke, and other cardiovascular diseases, making it a public health concern [3]. Hypertension is classified into five grades, each corresponding to a different severity level. Table 1 shows blood pressure and hypertension classes with grades [4].

Blood pressure is one of the most important vital signs to measure. It provides insight into an individual’s cardiovascular health by providing information about the amount of force exerted by the blood on the walls of the arteries. The traditional method for measuring blood pressure uses an inflatable cuff called the sphygmomanometer. The sphygmomanometer comprises an inflatable cuff, a bulb to inflate the cuff, a pressure gauge, and a stethoscope. The cuff is wrapped around the arm, and the bulb is squeezed to inflate the cuff. The pressure gauge monitors the pressure in the cuff, and the stethoscope listens for the sound of the blood as it rushes through the arteries. When the cuff pressure is increased beyond the systolic pressure, the sound of the turbulent blood can be heard through the stethoscope. The pressure at which the sound is heard is known as systolic pressure. As the cuff pressure is decreased, the sound disappears at the diastolic pressure.

Recently, cuffless blood pressure (CBP) measurement methods have been developed. These methods use optical, electrical, acoustic, and noninvasive technologies to measure blood pressure without an inflatable cuff [5,6,7,8]. Optical methods use fluctuations in light absorption due to the pulsatile blood flow in the arteries. In contrast, electrical methods rely on the electrical potentials generated by the pulsatile flow of the blood. Acoustic methods use acoustic pulse waves generated by the pulsatile flow of the blood, and other noninvasive methods use several technologies to measure blood pressure.

CBP measurement methods have several advantages over traditional cuff-based methods. First, they are noninvasive and thus do not require using an inflatable cuff with the associated discomfort. Second, they are more accurate and provide more detailed information about blood pressure. Third, they are faster since they do not require the time-consuming process of inflating and deflating the cuff. Finally, they are less prone to errors due to incorrect cuff placement and other user errors. Figure 1 shows cuffless blood pressure benefits and limitations [9,10,11].

The use of machine learning and deep learning algorithms in healthcare has been increasing rapidly in recent years as they are seen as essential tools to help improve patient outcomes and reduce costs. One particular application of this technology is in the field of CBP measurement. CBP measurement is a noninvasive technique that combines patient-specific physiological parameters to calculate an individual’s blood pressure without the traditional cuff. This technology has been growing in popularity due to its potential for providing accurate and reliable readings without the need for a physical cuff. In addition, it can be used in situations where traditional measurement techniques are not feasible or are impractical, such as in remote or rural areas. Machine learning and deep learning algorithms [12,13,14,15] have been used to develop automated systems for CBP measurement. There are several different machine learning and deep learning algorithms that can be used to develop automated CBP measurement systems. These include supervised learning algorithms such as support vector machines, decision trees, and random forests [16,17,18], and unsupervised learning algorithms such as K-means clustering and self-organizing maps [19,20,21]. Although each of these algorithms has its advantages and disadvantages, and the choice of algorithm will depend on the specific application and the data available, a combination of different algorithms may be needed for optimal results.

Photoplethysmography (PPG) is a noninvasive medical technique used to measure various physiological parameters such as blood pressure, heart rate, and respiration rate. It is based on the changes in light reflection from the skin that occur because of blood flow in the underlying tissue. It relies on photodetector sensors, which measure changes in the optical properties of the skin. PPG vital signals can be used to diagnose a variety of conditions, including hypertension, arrhythmia, and sleep apnea. For example, to measure blood pressure, PPG vital signals measure two variables: pulse wave amplitude and timing. Pulse wave amplitude is the magnitude of pulsation in the skin, while pulse timing is the duration of each pulse wave. As the blood pressure increases, the pulse wave amplitude increases, and the timing of the pulse wave will also change. By measuring these changes, blood pressure can be accurately calculated. Some articles showing the relationship between PPG and blood pressure are given [22,23,24,25,26].

This paper proposes a new continuous blood pressure (CBP) estimation application. Unlike previously published articles, CBP estimation uses time and chaotic attributes. Important information about the fluid mechanics of blood in the vessel is extracted from the vital PPG signal and used to estimate systolic and diastolic blood pressure values. PPG vital signals are preprocessed through base correction, filtering, and segmentation to prepare them for feature extraction. A total of 24 features, 17 time-domain features, and 7 chaotic-domain features were extracted and used for blood pressure estimation. Seven different regression models were used to make predictions, and the results were compared. A 10-fold cross-validation was used to increase the reliability of the predictions in the regression calculations.

This article has been organized as follows. Section 2 explains how PPG vital signals are used for CBP estimation. The next chapter shows the proposed method for our work. Section 4 explains the experimental results concerning the performance of the three methods in our work. Section 5 denotes the conducted works in the literature and gives the superiority of our work.

## 2. Material and Method

Blood pressure estimation is an important part of healthcare as it indicates overall health. Blood pressure is typically measured with a cuff; however, this is not always practical or comfortable. Therefore, CBP estimation has become an increasingly popular alternative. PPG vital signals are a type of technology that can be used for CBP estimation. PPG vital signals are noninvasive optical signals that measure the amount of light reflected from a person’s skin. These signals are generated by changing the amount of light entering the skin, and the changes in the amount of light are a reflection of the body’s blood flow. This blood flow can then be used to estimate the person’s blood pressure. PPG vital signals have several advantages over traditional cuff-based measurements. This makes them particularly useful for remote monitoring and measuring blood pressure during physical activities. In addition, PPG vital signals are more comfortable for patients, since no cuff is required. This is particularly beneficial for those who cannot tolerate or have difficulty using a cuff. To use PPG vital signals for CBP estimation, two sets of data must be collected: the PPG vital signals and the corresponding arterial pressure waveforms. The PPG vital signals can then be used to estimate the systolic and diastolic blood pressure values. This is carried out by using an algorithm to match the PPG vital signals to the arterial waveforms. The algorithm then calculates the blood pressure values based on the differences in the waveforms.

The study consists of four main parts. In the first part, preprocessing and segmentation processes are applied to the PPG signals in the dataset. The features determined in the second part are extracted from the PPG signal. In the third section, various regression models are trained with these features. In the fourth and last section, the trained models estimate the blood pressure as mmHg. The flow chart of the proposed model is given in Figure 2. Each section is explained above.

### 2.1. Datasets

The medical information mart for intensive care (MIMIC-III) database is a publicly available critical care database developed by the MIT Lab for Computational Physiology. It contains over 60,000 intensive care unit (ICU) admissions from approximately 41,000 patients collected between 2001 and 2012. The MIMIC-III database comprises multiple tables and contains over 60 million records. This includes patient demographics, laboratory results, vital signs, medications, diagnoses, procedures, and outcomes.

The MIMIC-III database contains five major tables: admissions, patients, ICU stays, physiological data, and clinical events. Physiological data and clinical events tables contain information about the patient’s vital signs, laboratory tests, and clinical events during their stay.

Vital signs are quantitative measurements that assess the function of the body’s organs and systems. Vital signs include heart rate, respiratory rate, blood pressure, temperature, and oxygen saturation. In MIMIC-III, the vital signs (ECG, PPG, ABP, etc.) are collected at regular intervals and stored in a database for further analysis. The data are also used as a tool for predictive analytics and machine learning applications. Figure 3 shows ECG, PPG, and ABP vital signals.

There was a negative correlation between the PPG and atrial blood pressure signals (ABP). El-Hajj et al. described and proved the correlation between the PPG and BP signals in detail [6,13,23]. Based on this information, we have performed experiments to show the relationship between the PPG signals and BP signals in our paper. As a result of the statistical studies, it was determined that there is a correlation coefficient of “−0.279” between the PPG signal and the ABP signal. Figure 4 shows the normalized PPG signals, the normalized ABP signals, and the negative graph between the PPG signals and ABP signals. Therefore, in this paper, we have used the PPG signals to predict the SBP and DBP values using machine learning methods with statistical results from the given figures.

### 2.2. Preprocessing Signals

Baseline wander is a preprocessing technique that normalizes and removes unwanted noise from physiological signals. It is essential in processing PPG signals, which are used in medical diagnostics, as the nature of the signal often contains abrupt changes, which can result in data misinterpretation. The median filter is one of the most commonly used methods for the baseline correction of PPG signals. The filter replaces each data point with the median of the points surrounding it. This smooths out any sudden changes in the signal, which can be caused by external noise or artifacts, such as body movements. This technique helps reduce the noise and artifacts, allowing for more accurate signal interpretation. The median filter is generally considered to be a better option than other types of filters, as it is less prone to overfitting and does not distort the original signal too much. This makes it more suitable for analyzing PPG signals, as it is important to maintain the signal’s integrity.

A low-pass filter is a type of filter used to remove higher-frequency components from a signal, allowing only the lower frequencies to pass through. It is commonly used to smooth out signals such as those derived from physiological signals, including PPG signals, which are used to measure physiological parameters such as heart rate and blood pressure. Low-pass filters are useful for filtering out noise or unwanted artifacts from PPG signals. They can be used to reduce the effects of high-frequency noise from environmental sources, such as electrical interference from other machines. Low-pass filters also help to reduce motion artifacts from patient movements, which can be particularly problematic for PPG signals. They can also help to reduce the effects of baseline wander, which is the slow drift of the baseline of the PPG signal. In the case of PPG signals, a low-pass filter with a cutoff frequency of 5 Hz can be used to remove any high-frequency noise that is present in the signal. Similarly, for ECG signals, a low-pass filter with a cutoff frequency of 40 Hz can be used to reduce noise.

### 2.3. Feature Extraction

The photoplethysmograph (PPG) signal is a noninvasive optical technique for measuring the changes in the blood volume in vessels and has been used as an alternative for measuring CBP. This technique uses time-domain and chaotic feature extraction to acquire the correct data. PPG signal analysis extracts features from the signal to measure the CBP accurately. Time-domain features are essential for cuffless blood pressure estimation because they give an insight into the dynamic behavior of the cardiovascular system. These features can measure how the body responds to changes in the external environment, such as physical activity and the intake of certain medications. Time-domain features also provide valuable information regarding the current blood pressure level and potential risk factors such as hypertension and cardiac arrhythmias. Time-domain features are essential for cuffless blood pressure estimation because they can help to distinguish between different pressure levels. Time-domain features are also helpful in detecting any conditions that may require further investigation. For example, a patient with high mean arterial pressure (MAP) may have an underlying condition, such as arteriosclerosis. Time-domain features can also be used to monitor individuals over time, which can help to detect any changes in the cardiovascular system. Time-domain features are essential for cuffless blood pressure estimation because they give us information about the cardiovascular system. They can help differentiate between different pressure levels, detect abnormalities, and monitor for changes over time.

Furthermore, they can give us insight into the dynamic behavior of the cardiovascular system, which can lead to a better understanding of the relationship between the cardiovascular system and external factors. Time-domain analysis extracts features from PPG signals such as Willison amplitude, signal variance, root mean square, log detector, etc. These features monitor blood pressure and analyze the PPG signals [8,22].

Chaotic feature extraction analyzes PPG signals and measures CBP [8,22]. Chaotic features provide a way to capture information about the cardiovascular system in a way that can be used to estimate blood pressure accurately. Chaotic features are used because of their ability to capture subtle changes in cardiovascular function that are difficult to detect with traditional methods. The chaotic features can detect and describe the complex, nonlinear dynamics of the cardiovascular system. They are able to capture features of the cardiovascular system that are not easily detected by traditional methods, such as the effect of respiration on blood pressure. This information can accurately estimate blood pressure without needing a cuff. Overall, chaotic features are used in cuffless blood pressure estimation because of their ability to capture subtle changes in cardiovascular function, increased accuracy and reliability, and cost-effectiveness. The use of chaotic features in cuffless blood pressure estimation is important for providing accurate and reliable estimates of blood pressure without the need for a traditional cuff. Entropy calculations are used to measure the randomness and complexity of the PPG signal, while the fractal dimension is used to measure the self-similarity of the signal. Time-domain and chaotic feature extraction accurately measure the CBP using PPG signals. These features monitor and analyze the PPG signal, providing a reliable and noninvasive method for measuring CBP. In the study, 24 features were extracted from each PPG segment. While 17 of them are time-domain features, 7 of them are chaotic features. Table 2 shows the extracted feature and its formulas.

### 2.4. Regression Models

One approach to predicting blood pressure from PPG signals is to use regression models. Regression models use data points to estimate a linear or nonlinear relationship between a dependent variable and one or more independent variables. In this context, the dependent variable is the blood pressure, and the independent variables are the PPG signals. Using regression models, it is possible to build a model that can accurately predict blood pressure from PPG signals. In addition, recent developments in machine learning algorithms have enabled PPG signals to predict blood pressure accurately. By using supervised learning algorithms—such as regression models—researchers have built models that can accurately estimate blood pressure levels from PPG signals. This can revolutionize blood pressure monitoring, allowing continuous real-time monitoring without invasive methods.

The advantage of using regression models for predicting blood pressure from PPG signals is that they can be tailored to the individual, allowing for a more accurate prediction. Additionally, regression models are relatively easy to use and interpret, making them highly accessible to researchers in the field.

This paper used different tools to observe the effect of regression tools on the problem. The regression models used are listed in Table 2. A 10-fold cross-validation was used in training the models. Using this method, the entire data set was divided into 10 parts. Then, models were trained and tested by turns, with 9 training datasets and 1 test dataset. Table 3 shows the regression models used.

#### 2.4.1. Linear Regression

Linear regression is a regression model used for linear and continuous variables. It is generally used to predict the value of one variable with another variable. The basic model is given in Equation (1) [27].
(1)$$y_{i}=\beta_{0}+\beta_{1}x_{i1}+\beta_{2}x_{i2}+\ldots +\beta_{n}x_{in}+\varepsilon_{i}$$

Here, y is the predicted variable and x is the predictive variable. β is the coefficient of the predictor variable. ε represents the fixed error of the model.

#### 2.4.2. Robust Linear Regression

Robust linear regression is a variant of linear regression. Robust linear regression is less sensitive to outliers [28].

#### 2.4.3. Rational Quadratic Gaussian Process Regression

Gaussian process regression is a Bayesian regression approach and is a nonparametric method. The Bayesian approach extracts a probability distribution over all possible values. In general, Bayes’ rule is expressed by Equation (2) [29].
(2)$$P\left(w|y,X\right)=\frac{P\left(y|X,w\right)P\left(w\right)}{P\left(y|X\right)}$$

Let $\left\{\left(x_{i},y_{i}\right);\;i=1,2,\ldots ,n\right\}$ be data. Where $x_{i}\epsilon R^{d}$ and $y_{i}\epsilon R$. The linear regression model is $y=x^{T}\beta +\varepsilon$.

It is commonly used to describe the statistical covariance at two points that are x units apart. The advantage of the rational quadratic GPR algorithm is that it is less likely to generate errors in large clusters [25]. The Rational GPR model becomes Equation (3).
(3)$$k\left(x_{i},x_{j}|\theta \right)=\sigma_{f}^{2}\left(1+\frac{r^{2}}{2\alpha \sigma_{1}^{2}}\right)$$
where $r=\sqrt{{\left(x_{i}-x_{j}\right)}^{T}\left(x_{i}-x_{j}\right)}$, θ is posterior estimates, $\sigma_{f}^{2}$ is signal variance, and α is a parameter of the covariance.

#### 2.4.4. Square Exponential Gaussian Process Regression

The square exponential Gaussian process is a radial basis function. It is essential to square the Euclidean distance. Square exponential Gaussian process regression formulas are given in Equation (4) [30].
(4)$$k\left(x_{i},x_{j}|\theta \right)=\sigma_{f}^{2}exp\left[-\frac{1}{2}\frac{{\left(x_{i}-x_{j}\right)}^{T}\left(x_{i}-x_{j}\right)}{\sigma_{1}^{2}}\right]$$

#### 2.4.5. Matérn 5/2 Gaussian Process Regression

The Matérn 5/2 kernel takes spectral densities and creates Fourier RBF kernel transforms. Functions in Matérn 5/2 |ν − 1| can be differentiated once, and the hyperparameter ν can control the degree of smoothness. The formula is shown by Equation (5) [30].
(5)$$k\left(x_{i},x_{j}|\theta \right)=\sigma_{f}^{2}\left(1+\frac{\sqrt{3r}}{\sigma_{1}}\right)\exp\left(-\frac{\sqrt{3r}}{\sigma_{1}}\right)$$

#### 2.4.6. Linear Support Vector Machine

Linear support vector machine allows the determination of the amount of acceptable error. A hyperplane is found to fit the data. The regression function is given by Equation (6) [31] for a linear functional dataset.
(6)$$\Phi \left(w,\xi^{*},\xi \right)=\frac{1}{2}{\left||w|\right|}^{2}+C\left[\overset{j}{\underset{i=1}{\sum }}\xi +\overset{j}{\underset{i=1}{\sum }}\xi^{*}\right]$$
where *C* is a penalty value and $\xi^{*},\xi$ are slack variables.

#### 2.4.7. Medium Gaussian Support Vector Machine

The SVM Gaussian kernel moves the data from the feature space to the higher dimensional kernel space and provides nonlinear separation in the kernel space. The Gaussian kernel function parameter k in Equation (7) takes the value $r=\sqrt{p}$ [32]. Here, *p* is the number of attributes.
(7)$$k\left(x_{i},x_{j}\right)=\exp\left(-\frac{||x_{i}-x_{j}||{}^{2}}{2\sigma^{2}}\right)=\exp\left(-r||x_{i}-x_{j}||{}^{2}\right)$$

## 3. Evaluated of Results

Applying a PPG signals regression model with CBP has been evaluated and showed promising results. The model could accurately predict blood pressure from the PPG signals, using both time-domain and chaotic features. The model’s evaluation showed that the predicted blood pressure accuracy was within a clinically acceptable range for most patients. The model’s accuracy was further improved by using the chaotic features, showing that the chaotic features could capture more information from the PPG signals and improve the accuracy of the predictions. Overall, the evaluation results showed that the model could accurately predict blood pressure from PPG signals with acceptable accuracy, showing the potential of using PPG signals to estimate blood pressure in clinical applications.

This paper proposes regression models for predicting systolic and diastolic blood pressure values from PPG signals. We first filter and clean the PPG signals from noise and artifacts. Then, we extract 24 features comprising the time domain and chaos from the PPG signals. These features are given as inputs to the regression models. After training and testing the regression models using 5-fold cross-validation, we can predict systolic and diastolic blood pressure values.

This paper uses the performance criteria to evaluate the proposed methods. These performance metrics are the mean square error (MSE), mean absolute error (MAE), root mean square error (RMSE), R^2^, mean absolute percentage error (MAPE), box plots of the predicted values, and the Bland–Altman plot. The performance metric formulas are given below:(8)$$MSE=\frac{1}{N}\sum_{i=1}^{N}{\left(y_{i}-{\hat{y}}_{i}\right)}^{2}$$
(9)$$RMSE=\sqrt{\frac{1}{N}\sum_{i=1}^{N}{\left(y_{i}-{\hat{y}}_{i}\right)}^{2}}$$
(10)$$MAPE=\frac{1}{N}\sum_{i=1}^{N}\left|\frac{y_{i}-{\hat{y}}_{i}}{y_{i}}\right|\times 100$$
(11)$$MAE=\frac{1}{N}\sum_{i=1}^{N}\left|y_{i}-{\hat{y}}_{i}\right|$$
(12)$$R^{2}=1-\frac{MSE}{MSE}$$

In these equations, $y_{i}\;$ is the actual value, ${\hat{y}}_{i}$ is the predicted value, and $\bar{y}$ is the mean of $y_{i}$. Moreover, N is the number of observations. The optimum value is 0 for MSE, MAPE, RMSE, and MAE. R^2^ takes a value of between –∞ and 1. Negative values indicate worse predictions.

Using the overall scores from all of the methods, we have given all of the obtained SBP and DBP prediction performance results values from three different methods, as shown in Table 4 and Table 5.

The comparison between regression model prediction and actual values for CBP prediction is important in determining the model’s efficacy. The correlation coefficient and Bland–Altman plots are ideal methods for assessing the accuracy of the regression models. The correlation coefficient measures the degree of linear correlation between two variables. At the same time, the Bland–Altman plot compares the differences between the predicted and the actual values over a range of data points. The correlation coefficient between the predicted and actual values for CBP prediction can be calculated using the Pearson correlation coefficient, which measures the degree of linear correlation between two variables. The correlation coefficient is a measure of the strength of the linear relationship between the two variables. A high correlation coefficient indicates that the two variables are strongly correlated, while a low correlation indicates that the two variables are weakly correlated. The Bland–Altman plot is an excellent way to compare the differences between the predicted and the actual values of CBP prediction. The Bland–Altman plot is a graphical representation of the differences between the two variables over a range of data points. The plot measures the variability in the differences between the two variables and is used to identify any outliers or patterns in the data. The Bland–Altman plot is a useful tool for assessing the accuracy of the regression models. Using these two methods to compare the regression model’s prediction and actual values makes it possible to assess the accuracy and reliability of the model. Correlation plots and the Bland–Altman plot are used for the best prediction model of SBP values using the Matérn 5/2 Gaussian process regression method, as shown in Figure 5.

Correlation plots and the Bland–Altman plot were used for the best prediction model of DBP values using the rational quadratic Gaussian process regression method, as shown in Figure 6.

The time series changes in the actual and estimated (Matérn 5/2 Gaussian process regression method) SBP values are shown in Figure 7. It can be seen that the SBP estimations made with the Matérn 5/2 Gaussian process regression method were successful. It has been observed that the performance of the predictions decreased with sudden increases and decreases in SBP.

The time series changes in the actual and estimated (rational quadratic Gaussian process regression method) DBP values are shown in Figure 8. It can be seen that the DBP estimations made with the rational quadratic Gaussian process regression method were successful. However, it has been observed that the performance of the predictions decreased with sudden increases and decreases in SBP.

## 4. Discussion

This paper presents a novel CBP prediction method based on regression models. As a noninvasive alternative to traditional methods of measuring blood pressure, this method is of particular importance for those with existing medical conditions or for those who are in remote locations. This paper details the preprocessing, feature extraction, and regression models used in the prediction process. This paper also demonstrates the performance of the regression methods used in the prediction process using the mean absolute error (RMSE, R^2^, MSE, and MAE) as the evaluation metric. The MAE values obtained for systolic and diastolic blood pressure are 3.073 and 1.721, respectively. The MAE values obtained by the authors are comparable to those obtained in other studies using the same dataset. In addition, this paper provides a clear description of the preprocessing, feature extraction, and regression models used in the prediction process—a compelling case for the efficacy of our proposed method. Table 6 shows a performance comparison of the conducted works in the literature and our proposed method for predicting SBP and DBP concerning the obtained MAE values.

Blood pressure estimation is an essential medical task studied extensively over the past few decades. In particular, the development of cuffless blood pressure estimation (CBPE) algorithms has enabled accurate and continuous blood pressure monitoring without needing any external device. While deep learning approaches have been successfully applied to this task, they have drawbacks. This article discusses the advantages of using regression models for CBPE over deep learning models. The first advantage of using regression models for CBPE is the model’s simplicity. Regression models are based on linear models and are easy to understand, interpret, and implement. They also require fewer parameters to be tuned, making them less computationally expensive compared to deep learning models.

Furthermore, they are less prone to overfitting, which can be a problem with deep learning models. Second, regression models can capture the underlying relationship between the input and output variables more accurately than deep learning models. Regression models can identify nonlinear relationships between the variables, whereas deep learning models may struggle to capture these relationships accurately. As a result, regression models are less prone to making erroneous predictions than deep learning models. The third advantage of using regression models for CBPE is that they are easily scalable. They can be easily applied to larger datasets and used in various contexts and settings.

In contrast, deep learning models require a lot of data in order to be effective and are more difficult to scale up. Finally, regression models can be used to identify and diagnose errors in the CBPE system. This is because the model’s parameters are easy to interpret and can be used to identify the sources of errors and misclassifications. In contrast, deep learning models are more difficult to interpret and diagnose, making it difficult to identify errors.

In conclusion, regression models have many advantages over deep learning models regarding CBPE. They are simple to understand and interpret, require fewer parameters to be tuned, can capture nonlinear relationships between the variables, are easily scalable, and it is easier to diagnose errors in the CBPE system. For these reasons, regression models are a viable alternative for CBPE. As shown in Table 6, regression models produced advantageous MAE values of 3.069 and 1.721 for SBP and DBP, respectively, in predicting blood pressure, resulting in higher performance.

## 5. Conclusions

The CBP measurement method is highly accurate and is much more comfortable for the user than the traditional cuff method. It is also much more cost-effective, as purchasing an arm cuff is unnecessary. This method is also very convenient as it does not require physical contact between the user and the device and can be performed quickly and easily. As a result, the CBP measurement method is becoming increasingly popular among healthcare professionals and patients. This method is particularly beneficial for those unable to use a traditional arm cuff due to disability or other health reasons. In addition, this method is becoming increasingly accessible as technology becomes more widely available. Ultimately, the CBP measurement method can provide accurate and reliable blood pressure readings and is an excellent alternative to the traditional cuff method.

This study concludes that a novel CBP prediction method using regression models is viable for accurately predicting systolic and diastolic blood pressures. This method uses preprocessing feature extraction and regression models to predict blood pressure. The preprocessing involved baseline wandering, filtering, and segmentation. Feature extraction involved both time domain and chaotic features. The regression models used were Matérn 5/2 Gaussian process regression, rational quadratic Gaussian process regression, and five different models for SBP and DBP. The results of this study show that the novel CBP prediction method is a viable option for accurately predicting systolic and diastolic blood pressures. The Matérn 5/2 Gaussian process regression method had a mean absolute error of 3.073, and the rational quadratic Gaussian process regression method had a mean absolute error of 1.721. This shows that the novel CBP prediction method can predict systolic and diastolic blood pressures with an accuracy comparable to other methods. The results of this study have implications for the medical industry. This method could be used to accurately predict blood pressure without needing a cuff, reducing the cost of medical care, and allowing patients to monitor their blood pressure more efficiently and accurately at home. In addition, this method could be used to monitor the blood pressure of patients with hypertension or hypotension, allowing for more accurate diagnoses and treatments. This study demonstrates that the novel CBP prediction method using regression models is viable for accurately predicting systolic and diastolic blood pressures. The results show that this method has a level of accuracy comparable to other methods and can be used to reduce the cost of medical care and more accurately diagnose and treat patients with hypertension or hypotension. This method is a promising step forward for the medical industry and could lead to more accurate and cost-effective treatments for patients.

## Figures and Tables

**Figure 1 diagnostics-13-01278-f001:**
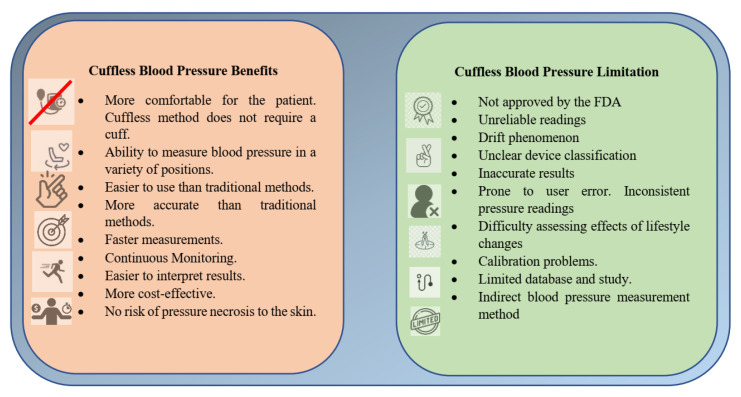
Benefits and limitations of cuffless blood pressure (CBP) measurement.

**Figure 2 diagnostics-13-01278-f002:**
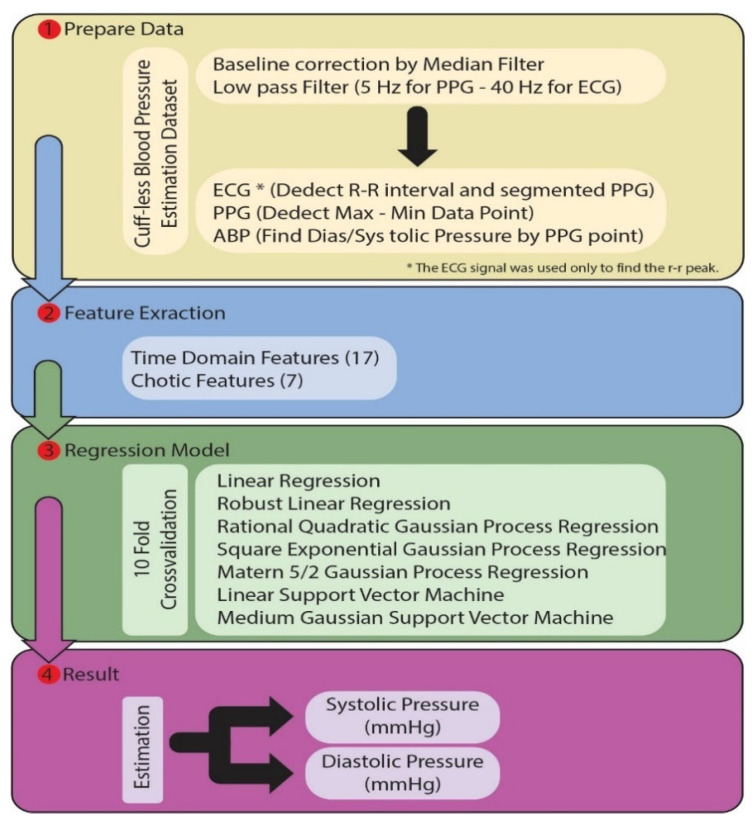
The flow chart of the proposed model for predicting SBP and DBP values.

**Figure 3 diagnostics-13-01278-f003:**
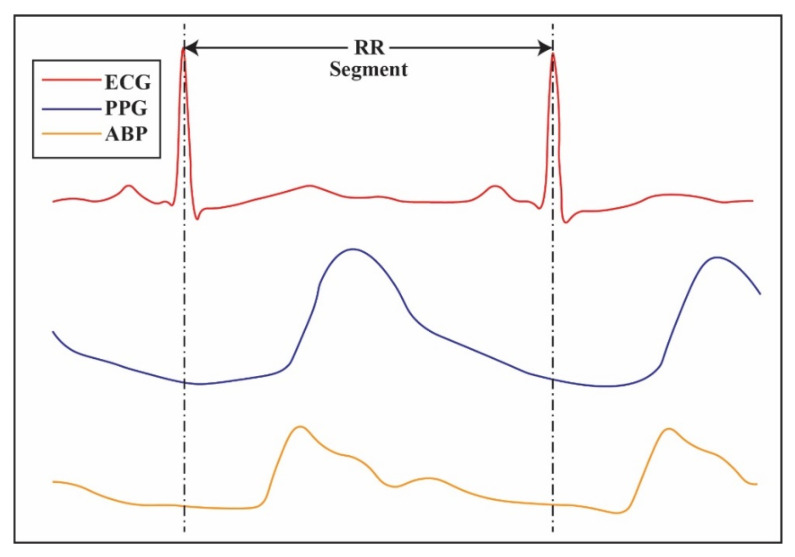
PPG, ECG, and ABP signals in MIMIC-III datasets.

**Figure 4 diagnostics-13-01278-f004:**
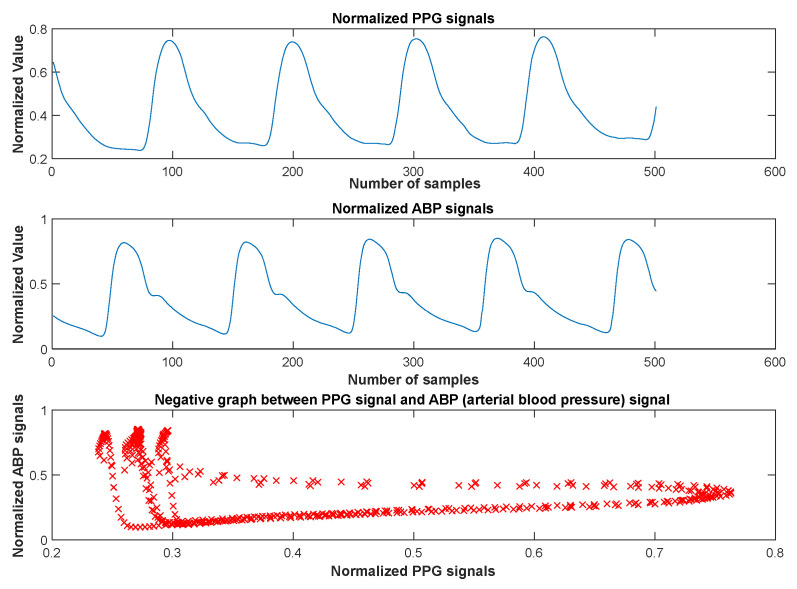
The normalized PPG signals, the normalized ABP signals, and the negative graph between the PPG signals and ABP signals. (The correlation coefficient between normalized PPG signals and normalized ABP signals is −0.279).

**Figure 5 diagnostics-13-01278-f005:**
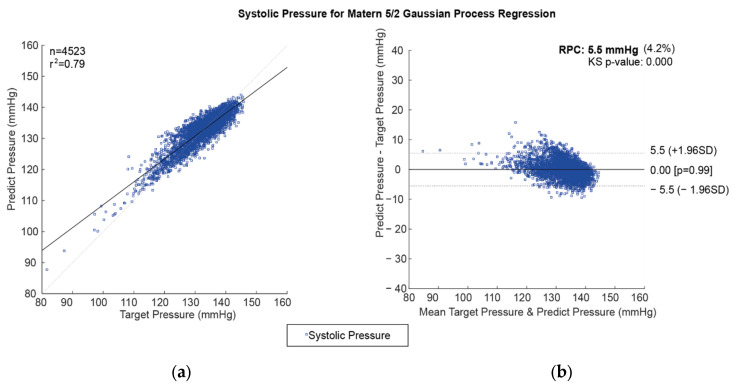
Predicted SBP for the Matérn 5/2 Gaussian process regression method: (**a**) the plot of the correlation and (**b**) the Bland–Altman Plot.

**Figure 6 diagnostics-13-01278-f006:**
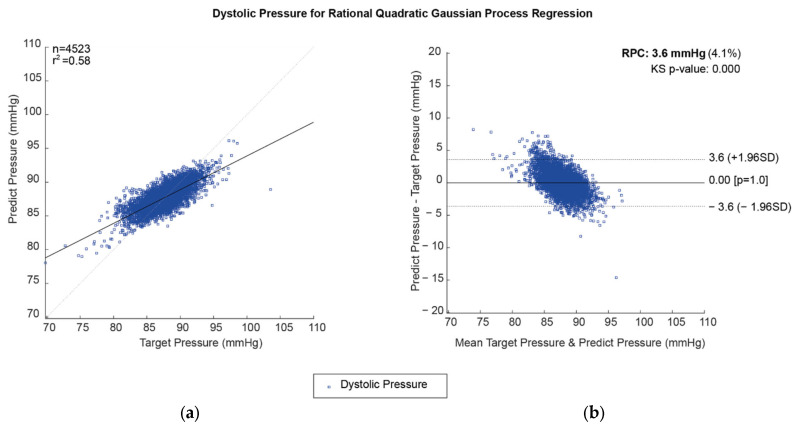
Predicted DBP for the rational quadratic Gaussian process regression method: (**a**) correlation plot and (**b**) the Bland–Altman Plot.

**Figure 7 diagnostics-13-01278-f007:**
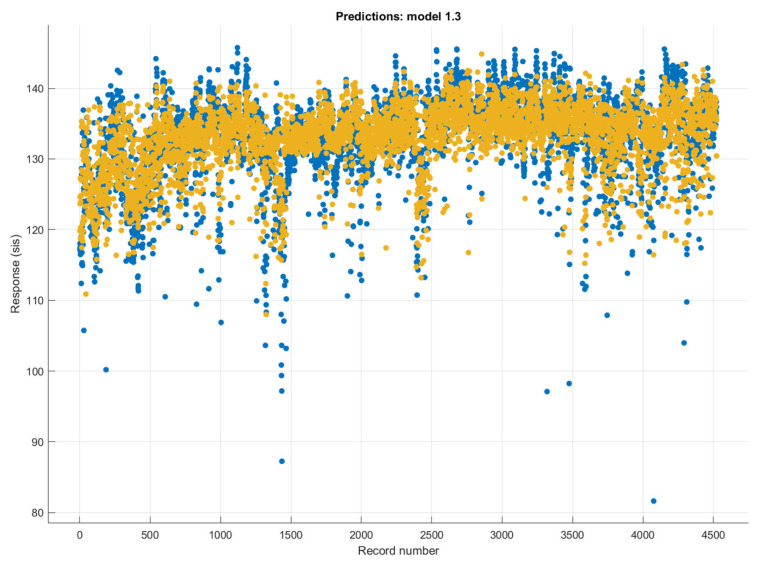
The target and predicted values in the prediction of SBP values using the Matérn 5/2 Gaussian process regression method (blue mark: actual SBP values; orange mark: the predicted SBP values).

**Figure 8 diagnostics-13-01278-f008:**
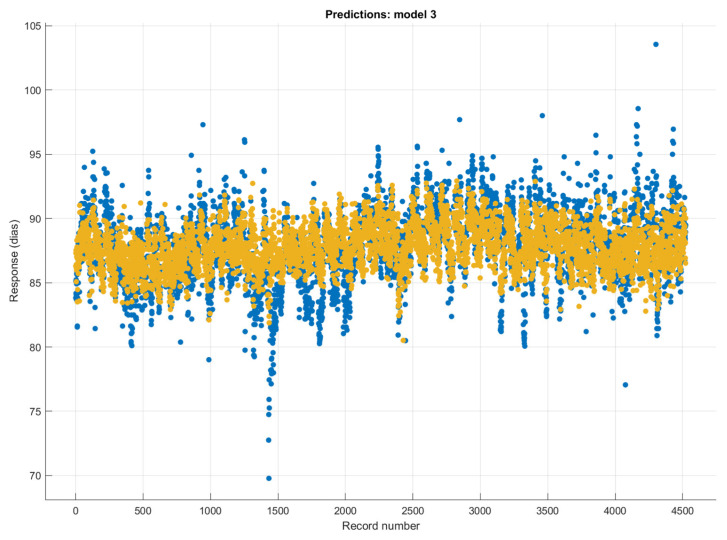
The target and predicted values in the prediction of DBP values using the rational quadratic Gaussian process regression method (blue mark: actual DBP values; orange mark: the predicted DBP values).

**Table 1 diagnostics-13-01278-t001:** Classification of blood pressure levels of the British Hypertension Society.

Category	Systolic Blood Pressure (mmHg)	Diastolic Blood Pressure (mmHg)
Blood pressure		
Optimal	<120	<80
Normal	<130	<85
High normal	130–139	85–89
Hypertension		
Grade 1 (Mild)	140–159	90–99
Grade 2 (Moderate)	160–179	100–109
Grade 3 (Severe)	≥180	≥110
Isolated systolic hypertension		
Grade 1	140–159	<90
Grade 2	≥160	<90

**Table 2 diagnostics-13-01278-t002:** The extracted features from PPG signals in the prediction of blood pressure.

Number of Features	Domain Information	Name of the Feature in the Dataset	Explanation of the Feature
1	Time	Enhanced Mean Absolute Value	$EMAV=\frac{1}{L}\overset{L}{\underset{i=1}{\sum }}\left|{\left(x_{i}\right)}^{P}\right|$
2	Time	Enhanced Wavelength	$EML=\overset{L}{\underset{i=2}{\sum }}\left|{\left(x_{i}-x_{\left(i-1\right)}\right)}^{P}\right|$
3	Time	Mean Absolute Value	$MAV=\frac{1}{L}\overset{L}{\underset{i=1}{\sum }}\left|x_{i}\right|$
4	Time	Wavelength	$w_{L}=\overset{L}{\underset{i=2}{\sum }}\left|x_{i}-x_{\left(i-1\right)}\right|$
5	Time	Zero Crossing	$zC=\overset{L-1}{\underset{i=1}{\sum }}f\left(x_{i}\right)$
6	Time	Slope Sign Change	$SSC=\overset{L-1}{\underset{i=2}{\sum }}f\left(x_{i}\right)$
7	Time	Root Mean Square	$RMS=\sqrt{\frac{1}{L}\overset{L}{\underset{i=1}{\sum }}{\left(x_{i}\right)}^{2}}$
8	Time	Average Amplitude Change	$AAC=\frac{1}{L}\overset{L-1}{\underset{i=1}{\sum }}\left|x_{i+1}-x_{i}\right|$
9	Time	Difference Absolute Standard Deviation Value	$DASDV=\sqrt{\frac{\sum_{i=1}^{L-1}{\left(x_{i+1}-x_{i}\right)}^{2}}{L-1}}$
10	Time	Log Detector	$LD=\exp\left(\frac{1}{L}\overset{L}{\underset{i=1}{\sum }}\log\left(\left|x_{i}\right|\right)\right)$
11	Time	Modified Mean Absolute Value 1	$MMAV=\frac{1}{L}\overset{L}{\underset{i=1}{\sum }}w_{i}\left|x_{i}\right|$
12	Time	Modified Mean Absolute Value 2	$MMAV2=\frac{1}{L}\overset{L}{\underset{i=1}{\sum }}w_{i}\left|x_{i}\right|$
13	Time	Myopulse Percentage Rate	$MYOP=\frac{1}{L}\overset{L}{\underset{i=1}{\sum }}f\left(x_{i}\right)$
14	Time	Simple Square Integral	$SSI=\overset{L}{\underset{i=1}{\sum }}{\left(x_{i}\right)}^{2}$
15	Time	Variance of Signal	$VAR=\frac{1}{L-1}\overset{L}{\underset{i=1}{\sum }}{\left(x_{i}\right)}^{2}$
16	Time	Willison Amplitude	$WA=\overset{L-1}{\underset{i=1}{\sum }}f\left(x_{i}\right)$
17	Time	Maximum Fractal Length	$MFL=\log_{10}\left(\sqrt{\overset{L-1}{\underset{i=1}{\sum }}{\left(x_{i+1}-x_{i}\right)}^{2}}\right)$
18	Chaotic	Sample Entropy	$SampEn\left(m,r,N\right)=-\log\left(\overset{N-m}{\underset{i=1}{\sum }}A_{i}/\overset{N-m}{\underset{i=1}{\sum }}B_{i}\right)$
19	Chaotic	Approximate Entropy	$ApEn\left(m,r,N\right)=-\frac{1}{N-m}\overset{N-m}{\underset{i=1}{\sum }}\log\left(\frac{Ai}{B_{i}}\right)$
20	Chaotic	Fuzzy Entropy	$E\left(A\right)=-k\left\{\mathrm{mA}\left(\mathrm{xi}\right)\mathrm{logmA}\left(\mathrm{xi}\right)+\left(1-\mathrm{mA}\left(\mathrm{xi}\right)\right)\log\left(1-\mathrm{mA}\left(\mathrm{xi}\right)\right)\right\}$
21	Chaotic	Shannon Entropy	$H\left(\alpha \right)=-\underset{1}{\sum }P_{i}*\log p_{i}$
22	Chaotic	Permutation Entropy	$H\left(n\right)=-\overset{n!}{\underset{i}{\sum }}p\left(\pi_{i}\right)\log p\left(\pi_{i}\right)$
23	Chaotic	Higuchi Fractal Dimension	$l\left(k\right)=\frac{\left(\sum_{i=1}^{\left[N-m/k\right]}\left|x\left(m+i_{k}\right)-x\left(m+{\left(i-1\right)}_{k}\right)\right|\left(N-1\right)\right)}{\left(\frac{N-m}{k}\right)k}$
24	Chaotic	Katz Fractal Dimension	$FD_{katz-norm}=\frac{\log_{10}\left(L∕a\right)}{\log_{10}\left(d∕a\right)}=\frac{\log_{10}\left(n\right)}{\log_{10}\left(\frac{d}{L}\right)+\log_{10}\left(n\right)}$

**Table 3 diagnostics-13-01278-t003:** List of regression models.

No	Model Name
1	Linear regression
2	Robust linear regression
3	Rational quadratic Gaussian process regression
4	Square exponential Gaussian process regression
5	Matérn 5/2 Gaussian process regression
6	Linear support vector machine
7	Medium Gaussian support vector machine

**Table 4 diagnostics-13-01278-t004:** The obtained SBP prediction metric rates in our study using three different machine learning methods.

Regression Methods	RMSE	R^2^	MSE	MAE
Linear regression	4.893	0.470	20.195	3.272
Robust linear regression	4.557	0.450	20.767	3.227
Rational quadratic Gaussian process regression	4.279	0.520	18.318	3.069
Square exponential Gaussian process regression	4.305	0.510	18.318	3.090
Matérn 5/2 Gaussian Process Regression	4.277	0.52	18.297	3.073
Linear support vector machine	4.527	0.46	20.494	3.267
Medium Gaussian support vector machine	4.399	0.490	19.353	3.107

**Table 5 diagnostics-13-01278-t005:** The obtained DBP prediction metric rates in our study using three different machine learning methods.

Regression Methods	RMSE	R^2^	MSE	MAE
Linear regression	2.463	0.228	6.071	1.872
Robust linear regression	2.244	0.386	5.035	1.861
Rational quadratic Gaussian process regression	2.303	0.330	5.306	1.721
Square exponential Gaussian process regression	2.325	0.310	5.409	1.736
Matérn 5/2 Gaussian process regression	2.309	0.320	5.335	1.724
Linear support vector machine	2.514	0.200	6.321	1.857
Medium Gaussian support vector machine	2.328	0.310	5.420	1.732

**Table 6 diagnostics-13-01278-t006:** The performance comparison of the conducted works in the literature and our proposed method in predicting SBP and DBP concerning the obtained MAE values.

Compared Methods	Systolic MAE (mmHg)	Diastolic MAE (mmHg)	Ref.
Generalized deep neural network model	3.21	2.23	[33]
Spectro-temporal deep neural network	9.43	6.88	[34]
Fully convolutional neural networks	5.73	3.45	[35]
Machine learning model	9.54	5.48	[36]
Deep learning model	4.51	2.6	[13]
Regression by MARS (dynamical approach)	7.83	4.86	[37]
Tree-based pipeline optimization tool	6.52	4.19	[38]
CNN representations of PPG	4.48	2.19	[39]
Matérn 5/2 Gaussian process regression and feature extraction	3.073	1.724	Proposed Methods
Rational quadratic Gaussian process regression and feature extraction	3.069	1.721	

## Data Availability

We have provided our dataset in our paper. If you use our dataset, please cite our paper in your paper. The dataset explanation is here: There are 9 .mat files in the folder, each of which belongs to a different person. Each .mat file contains data named “Raw Data” and “z”. “Raw Data” is the state of personal EEG signals after passing through a 50 Hz Notch filter and 1–12 Band-pass filter. “z” indicates the indexes of the S tags belonging to that person on the “Raw Data”. The link to the dataset is as follows: https://drive.google.com/drive/folders/1BwsRNND79onkooH1jnjQBBLkGoOJOPcX?usp=sharing (accessed on 10 March 2023).

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
