# Peer review of "A Novel Cuffless Blood Pressure Prediction: Uncovering New Features and New Hybrid ML Models"

_diagnostics, 2023, doi:10.3390/diagnostics13071278_

Round 1

Reviewer 1 Report

Present manuscript, “A novel Cuffless Blood Pressure Prediction: Uncovering New Features and New Hybrid ML models” proposes  feature based regression methods for predicting cuffless blood pressure, and utilizes PPG data available on MIMIC-III database for the cuffless Blood pressure prediction.

Comment 1: Introduction is mainly on Blood pressure its ranges and impact, which can be made concise, and has very less technical information about the cuffless measurement methods their specific advantages and limitations.

Comment 2: Need to add the concept how PPG relates to Blood pressure, present study does not have any clinical trial or corelation study with respect to this.

Comment 3: There is need to add criterion for selecting the features.

Comment 4: Need to add the algorithm, and flow chart, specificity and sensitivity of the method proposed in analyzing Cuffless Blood Pressure.

Author Response

Comment 1: Introduction is mainly on Blood pressure its ranges and impact, which can be made concise, and has very less technical information about the cuffless measurement methods their specific advantages and limitations.

Response: Thank you for your valuable comment. We have corrected this sentence in the revised paper.  

Based on your valuable comments, we have added some sentences to the revised paper:

CBP measurement methods have several advantages over traditional cuff-based methods.  First, they are non-invasive and thus do not require using an inflatable cuff and the associated discomfort.  Second, they are more accurate and provide more detailed information about blood pressure.  Third, they are faster since they do not require the time-consuming process of inflating and deflating the cuff.  Finally, they are less prone to errors due to incorrect cuff placement and other user errors. Figure 1 shows cuffless blood pressure's benefits and limitations [9-11].

Figure 2: Benefits and limitations of cuffless blood pressure (CBP)

Comment 2: Need to add the concept how PPG relates to Blood pressure, present study does not have any clinical trial or corelation study with respect to this.

Response: Thank you for your valuable comment. We have added some sentences regarding “how PPG relates to Blood pressure” in the revised paper. We have explain them in the revised paper.

The added sentences are given as follows:

Photoplethysmography (PPG) is a non-invasive medical technique used to measure various physiological parameters such as blood pressure, heart rate, and respiration rate. It is based on the changes in light reflection from the skin that occur because of blood flow in the underlying tissue. It relies on photodetector sensors, which measure changes in the optical properties of the skin. PPG vital signals can be used to diagnose a variety of conditions, including hypertension, arrhythmia, and sleep apnea. For example, to measure blood pressure, PPG vital signal measures two variables: pulse wave amplitude and timing. Pulse wave amplitude is the magnitude of pulsation in the skin, while pulse timing is the duration of each pulse wave. As the blood pressure increases, the pulse wave amplitude increases, and the timing of the pulse wave will also change. By measuring these changes, blood pressure can be accurately calculated. Some articles showing the relationship between PPG and blood pressure are given [22-26].

Comment 3: There is need to add criterion for selecting the features.

Response: Thank you for your valuable comment.  We have explained in the relevant subsection in the revised paper. Also, in our paper, we have not used any feature selection algorithm. Totally, we have used 24 features including the time and chaotic features obtained from the PPG signals. We chose these features because the features that best characterize the PPG signal are time and chaotic features.

The added sentences are given as follows:

The photoplethysmograph (PPG) signal is a non-invasive optical technique for measuring the changes in the blood volume in vessels and has been used as an alternative for measuring CBP. This technique uses time domain and chaotic feature extraction to acquire the correct data. PPG signal analysis extracts features from the signal to measure the CBP accurately. Time domain features are essential for cuffless blood pressure estimation because they give us insight into the dynamic behavior of the cardiovascular system. These features can measure how the body responds to changes in the external environment, such as physical activity and the intake of certain medications. Time domain features also provide valuable information regarding the current blood pressure level and potential risk factors such as hypertension and cardiac arrhythmias. Time domain features are essential for cuffless blood pressure estimation because they can help to distinguish between different pressure levels. Time domain features are also helpful in detecting any conditions that may require further investigation. For example, a patient with high mean arterial pressure (MAP) may have an underlying condition, arteriosclerosis. Time domain features can also be used to monitor individuals over time, which can help to detect any changes in the cardiovascular system. Time domain features are essential for cuffless blood pressure estimation because they give us information about the cardiovascular system. They can help differentiate between different pressure levels, detect abnormalities, and monitor for changes over time.

Furthermore, they can give us insight into the dynamic behavior of the cardiovascular system, which can lead to a better understanding of the relationship between the cardiovascular system and external factors. Time domain analysis extracts features from PPG signals such as Willison amplitude, signal variance, root means square, log detector, etc. These features monitor blood pressure and analyze the PPG signals [8,22].  

Chaotic feature extraction analyzes PPG signals and measures CBP [8,22]. Chaotic features provide a way to capture information about the cardiovascular system in a way that can be used to estimate blood pressure accurately. Chaotic features are used because of their ability to capture subtle changes in cardiovascular function that are difficult to detect with traditional methods. The chaotic features can detect and describe the complex, non-linear dynamics of the cardiovascular system. They are able to capture features of the cardiovascular system that are not easily detected by traditional methods, such as the effect of respiration on blood pressure. This information can accurately estimate blood pressure without needing a cuff. Overall, chaotic features are used in cuffless blood pressure estimation because of their ability to capture subtle changes in cardiovascular function, increased accuracy and reliability, and cost-effectiveness. The use of chaotic features in cuffless blood pressure estimation is important for providing accurate and reliable estimates of blood pressure without the need for a traditional cuff. Entropy calculations are used to measure the randomness and complexity of the PPG signal, while the fractal dimension is used to measure the self-similarity of the signal. Time domain and chaotic feature extraction accurately measure the CBP using PPG signals. These features monitor and analyze the PPG signal, providing a reliable and non-invasive method for measuring CBP. In the study, 24 features were extracted from each PPG segment. While 17 of them are time domain features, 7 of them are chaotic features. Table 2 shows the extracted feature and its formulas.

Comment 4: Need to add the algorithm, and flow chart, specificity and sensitivity of the method proposed in analyzing Cuffless Blood Pressure.

Response: Thank you for your valuable comment. In our paper, we have estimated the DBP and SBP values using the hybrid regression models. Our problem is the data estimation problem with respect to the machine learning perspective. So, since there is no the classification problem, we have not used the specificity and sensitivity values in the evaluation of our models.

We have used the MSE, R2, RMSE, and MAE as the performance criteria in our paper.

Reviewer 2 Report

Majid et al in this manuscript "A novel Cuffless Blood Pressure Prediction: Uncovering New Features 2 and New Hybrid ML models" have provided a novel approach for predicting cuffless blood pressure that can be used to develop more accurate models in the future.

This is well-written with supportive evidence. I am recommending this for acceptance in the present form.

Author Response

Reviewer 2:

Majid et al in this manuscript "A novel Cuffless Blood Pressure Prediction: Uncovering New Features 2 and New Hybrid ML models" have provided a novel approach for predicting cuffless blood pressure that can be used to develop more accurate models in the future.

This is well-written with supportive evidence. I am recommending this for acceptance in the present form.

Response: Thank you for your valuable comment. We have improved our paper according to the comments.

Round 2

Reviewer 1 Report

As suggested in the earlier review also, there is need to include  a co-relation study for PPG based Blood pressure measurement and the blood pressure measured by conventional sphygmomanometer and stethoscope method, i.e. clinical co-relation study is desired to claim the novelty of this Cuffless Blood Pressure Prediction method.

Author Response

Manuscript ID: diagnostics-2253312

Type of manuscript: Article- revised paper

Title: “A novel Cuffless Blood Pressure Prediction: Uncovering New Features and New Hybrid ML models”

Reviewer 2:

Comment:  As suggested in the earlier review also, there is need to include  a co-relation study for PPG based Blood pressure measurement and the blood pressure measured by conventional sphygmomanometer and stethoscope method, i.e. clinical co-relation study is desired to claim the novelty of this Cuffless Blood Pressure Prediction method.

Response: Thank you so much for your kind comment. We have given a response to your comment.

There was a negative correlation between the PPG and atrial blood pressure signals (ABP). El-Hajj et al. described and proved the correlation between the PPG and BP signals in detail [6, 13, 23]. Based on this information, we have done experiments to show the relationship between the PPG signals and BP signals in our paper. As a result of the statistical studies, it was determined that there is a correlation coefficient of “-0.279” between the PPG signal and the ABP signal. Figure 4 shows the normalized PPG signals, the normalized ABP signals, and the negative graph between the PPG signals and ABP signals. So, in this paper, we have used the PPG signals to predict the SBP and DBP values using machine learning methods with statistical results from the given figures.

Figure 1. The normalized PPG signals, the normalized ABP signals, and the negative graph between the PPG signals and ABP signals. (The correlation coefficient between normalized PPG signals and normalized ABP signals: -0.279)
